



# Evaluating the use of Aeolus satellite observations in the regional NWP model Harmonie-Arome

Susanna Hagelin[1], Roohollah Azad[2], Magnus Lindskog[1], Harald Schyberg[2], and Heiner Körnich[1]

[1]SMHI, Norrköping, Sweden
[2]Met Norway, Oslo, Norway

**Correspondence:** Susanna Hagelin (susanna.hagelin@smhi.se)

**Abstract.** The impact of using wind speed data from the Aeolus satellite in a limited area Numerical Weather Prediction (NWP) system is being investigated using the limited area NWP model Harmonie-Arome over the Nordic region. We assimilate the Horizontal Line of Sight (HLOS) winds observed by Aeolus using 3D-Var data assimilation for two different periods, one in Sept-Oct 2018 when the satellite was recently launched, and a later period in Apr-May 2020 to investigate the updated data

processing of the HLOS winds. We find that the quality of the Aeolus observations have degraded between the first and second experiment period over our domain. However observations from Aeolus, in particular the Mie winds, have a clear impact on the analysis of the NWP model for both periods whereas the forecast impact is neutral when compared against radiosondes. Results from evaluation of observation minus background and observation minus analysis departures based on Desroziers diagnostics show that the observation error should be increased for Aeolus data in our experiments, but the impact of doing so is small.

We also see that there is potential improvement in using 4D-Var data assimilation, which generate flow-dependent analysis increments, with the Aeolus data.

## 1 Introduction

It is well known that the quality of Numerical Weather Prediction (NWP) forecasts are dependent on the accuracy of the estimation of the initial state (Lorenz, 1965). The process of combining model information, in the form of a so called background,

with various types of observations for producing a model initial state is referred to as data assimilation. In particular the use of satellite radiances has been demonstrated to be very important for the quality of NWP (Geer and Coauthors, 2017). There are also satellite wind products, such as atmospheric motion vectors (AMV) derived from satellite radiances. However, there is clearly a lack of direct accurate wind speed observation for all layers of the atmosphere and available over all areas of globe. Existing observing system already provide valuable data but there are gaps in the coverage as identified by the World

Meteorological Organisation (WMO, Böttger et al. (2004)). Radiosonde locations are unevenly distributed, and usually only available twice per day and mainly over land areas, winds derived from satellite Atmospheric Motion Vectors (AMV) are only available where there are cloud and can only measure the wind speed at cloud top height. Data from aircraft and air traffic control systems can sample vertical sections, but only during take-off and landing, the rest of the data comes from the height of the flight level. There is a clear gap in data coverage over remote areas like the Pacific Ocean and over the poles.





The Aeolus satellite is a polar orbiting wind profiler and part of the European Space Agency's (ESA) Earth Explorer mission (Stoffelen et al., 2005; Martin et al., 2020). Since the launch on 22 August 2018 it is orbiting the Earth in a sun-synchronous orbit at 320 km height providing vertical wind speed profiles measured with a Doppler wind lidar. A Doppler wind lidar is an active instrument and it derives the wind speed measurements by detecting the Doppler shift in the backscatter from the onboard laser using an instrument called ALADIN (Atmospheric Laser Doppler Instrument, see Reitebuch et al. (2009)). The

wind speed observed by the satellite is perpendicular to the direction of travel, hereafter referred to as the HLOS (Horizontal Line of Sight) wind. This means that the wind speed measured by the Aeolus satellite is most impacted by winds in the east-west direction. The ALADIN instrument onboard the Aeolus satellite measures two modes of scattering, Rayleigh and Mie. The Rayleigh measurements are used in the clear atmosphere and are derived from the molecular backscatter from the atmosphere. They reach a higher altitude than the Mie winds but also have a larger uncertainty. The Mie winds rely on measuring the cloud

and aerosol backscatter.The Mie winds have a stronger signal than the Rayleigh ones, and thus Mie winds are more precise and can be derived with a higher vertical resolution. They are also more concentrated to the lower part of the atmosphere.

The Aeolus satellite is the first satellite-based Doppler wind lidar mission in the world and is demonstrating the potential of this technique for obtaining global information on the vertical distribution of the wind speed. In particular there is a need for high resolution wind speed information with accurate height assignment, so the wind shear, which is important both for

diagnosing turbulence and for predicting developing baroclinic weather systems, can be accurately taken into account by both regional and global NWP systems.

The Aeolus satellite has now been in orbit for over two years and has been tested extensively by many weather forecasting centres around the world and has been shown to improve the forecast at, for example, the European Centre for Medium-range Weather Forecasting (ECMWF)) (Rennie and Isaksen, 2020) Similar results have been seen in testing by Météo-France, DWD

(the German weather service) and the UK Met Office. Observations from the Aeolus satellite is now used in operational global models at ECMWF, DWD, Météo-France (Martin et al., 2020; Pourret and co authors, 2021) and the UK Met Office, though only the Mie observations are used at the UK Met Office (Halloran, 2020).

In this study we want to evaluate the suitability of using data from the Aeolus satellite in a limited area model. To our knowledge this is the first study investigating the quality and impact of Aeolus data using a km-scale limited area model (LAM)

using real Aeolus observations. Previous studies, for example Šavli et al. (2018) evaluated the use of pre-launch test data in a 15 km LAM using the WRF model (Weather Research and Forecasting, Skamarock et al. (2008)). We use the Harmonie-Arome model (Bengtsson et al., 2017) over the Nordic countries, covering the operational domain of MetCoOp (Meteorological Cooperation, Müller et al. (2017)). The use of Aeolus observations in limited area km-scale data assimilation differs in several aspects from global data assimilation. In particular, for a limited-area modelling system Aeolus data is available over the

domain only sometimes and the difference in spatial scales represented by the observation and by model, respectively, is in some aspects, relatively large. There are also particular operational constraints for km-scale limited area modelling, such as need for short latency of Aeolus observations. We will look at the overall impact of Aeolus data and as well as the impact of the Rayleigh and Mie observations.





## 2 Description of the NWP model and study choice

We use the Harmonie-Arome (Bengtsson et al., 2017) version (cy43) of the shared Aire Limitée Adaptation dynamique Developpement InterNational (ALADIN)- High Resolution Limited Area Model (HIRLAM) NWP system. The three main components of this system are surface data assimilation, upper-air data assimilation and the forecast model. Here we focus on the upper-air data assimilation that has been prepared for assimilation of Aeolus HLOS observations. The data assimilation is applied within a 3 h data assimilation window, in which a background state is combined with various types of observations to

obtain model initial states. All experiments are run over the MetCoOp domain, covering Norway, Sweden, Finland and Estonia using a 2.5 km grid size and 65 vertical levels, with a model top at approximately 10 hPa. The model domain is shown in Fig. **??**, which also the location of the available Aeolus observations. The lateral boundary conditions (LBCs) are provided by the deterministic forecast from the IFS run by ECWMF. These forecasts are launched every 6 h with a 1 h output frequency. In addition, to benefit from the high-quality large scale information from the ECMWF global forecasts in the regional MetCoOp

data assimilation, a spectral large scale mixing of the background state fields with the lateral boundary ECMWF IFS fields is applied.

In the main part of this paper we use the Harmonie-Arome (cy 43) standard data assimilation setup with a three-dimensional variational data assimilation (3D-Var). The data assimilation is using all available conventional observations, aircraft data and AMSU-A/MHS radiances from polar orbiting satellites NOAA18 and 19 and MetOp1 and 2 as well as Aeolus HLOS winds.

For the satellite radiance data a so-called variational bias correction (VarBC) based on ideas of Dee (2005); Dee and Uppala (2009) is applied. Background-error statistics are calculated from an ensemble of forecast differences (Berre, 2000; Brousseau et al., 2012). These are produced by Ensemble Data Assimilation experiments (EDA) with perturbed observations carried out with the Harmonie-Arome system applying ECMWF global EDA forecasts as lateral boundary conditions. Scaling is applied to the derived statistics in order to be in agreement with the amplitude of Harmonie-Arome + 3 h forecast errors. Background

as well as observation errors are assumed to have a Gaussian error distribution as characterized by their error covariances. Observation errors are assumed to be uncorrelated and their Gaussian distribution within the minimization of the cost function is represented by the error variances. Assumed observation error statistics for all types of observations, except Aeolus HLOS, are static and based on data assimilation studies. The observation handling main components are the observation operators, projecting the model state on the observed quantities, and the background check, rejecting observations which are assumed to

be affected by gross errors and identified by large observation minus background departures. In addition a thinning is applied to some spatially dense data (such as satellite radiances, and aircraft observations) in order to alleviate effects of spatially correlated observation errors not represented in the data assimilation.

The Aeolus HLOS observation operator H consists of a vertical interpolation to the level of the observation, followed by a projection of the model wind field on the horizontal line of sight from the observed position in the direction towards the

satellite. An Aeolus HLOS observation $y_i$, is rejected if it does not satisfy the following inequality:

$$([H(\mathbf{x}^b)]_i - y_i)^2 / \sigma_{b,i}^2 > L \times \lambda, \tag{1}$$



where $\lambda = 1 + \sigma_{o,i}^2/\sigma_{b,i}^2$, $\sigma_{o,i}$ is the observation error standard deviation, $\sigma_{b,i}$ is the background error standard deviation, $L$ is the rejection limit and $[H(\mathbf{x}^b)]_i$ denotes the projection of the model background state $x^b$ on observation $i$.

The Aeolus product used in this study is the so called L2B wind product which provides the HLOS wind speed. This is developed by ECMWF and KNMI and the rest of the Aeolus Data, Innovation, and Science Cluster (DISC) team under contract from ESA. The L2B processing also provide an estimate of the observation instrument noise and corrections for temperature and pressure dependencies of the Rayleigh winds using a priori information from the ECMWF model (Rennie and Isaksen, 2020).

The first set of experiments in this study for laser A was run from 14 September 2018 to 14 October 2018 as recommended by ESA. The ALADIN instrument showed some decay in laser energy over time, and this was a period ESA considered the data quality to be quite good.The Aladin instrument showed some decay in laser energy over time, and this was a period ESA considered the data quality to be quite good.

On a later stage in the mission ESA reconfigured ALADIN to use a second available laser, laser B, to improve the data quality, and we have run a second set of experiments focusing on the performance of laser B, starting 20 April 2020 and ending 19 May 2020. This period was chosen because the Aeolus data with corrections for the M1 mirror temperature bias (Martin et al., 2020) became available and the data should thus have a higher quality than for the previous weeks.

During the second period, Aeolus is used operationally by the ECMWF, so indirectly there will be some influence from Aeolus data in the LBCs for this set of experiments. For all experiments, both for the laser A and the laser B period, the same set of LBCs are used for all parallel experiments. Thus only the impact coming from the km-scale limited-area data assimilation of Aeolus data will be investigated in this study. To exploit the impact coming from the LBC and from the large-scale mixing from introduction of Aeolus data a coordinated experiment with ECMWF would have been needed, with ECMWF providing to sets of LBC data, with and without assimilation of Aeolus in the ECMWF global model. The use of same LBC data for both parallel experiments can therefore limit the potential impact we can see from the Aeolus data. The impact of LBC and large-scale mixing versus regional model data assimilation in Harmonie-Arome is discussed in Randriamampianina et al. (2020).

All experiment run the model every three hours with 3D-Var data assimilation. The model is running a 12 h forecast at the main cycles at synoptic times (00, 06, 12, 18 UTC) and catch-up runs of with a three hour forecast for the remaining cycles.

## 3 Characteristics of Aeolus data

The Aeolus satellite is a research satellite and the first satellite-based lidar mission in the world. The HLOS wind observations used in this study are L2B wind products provided by ECMWF and KNMI which are suitable for data assimilation in NWP models (Rennie and Isaksen, 2020). For each Aeolus measurements 20 laser pulses are accumulated corresponding to a horizontal resolution of approximately 2.9 km. The observations are then made by averaging up to 30 individual measurements for both Rayleigh and Mie channels which results in a horizontally averaged wind data of about 80 km. The higher signal-to-noise ratio observed for the Mie channel made it possible to have the horizontal integration length being decreased



to about 10 km after 5 March 2019. To avoid the systematic errors, corrections are made for the temperature and pressure dependence of Rayleigh data using a priori information from the ECMWF model (Dabas et al., 2008). Moreover, the wind data is retrieved in 24 bins in the vertical where the resolution varies from 0.25 km near the surface to 2 km at the higher levels.

The method used to derive the wind speed data from the raw measurements and the knowledge of how to best use the data in an NWP model is continually under development. Also the L2B processing software is continuously updated so there are

some differences in the processing of the HLOS data for our chosen periods, most notably the correction introduced for the orbital bias caused by the difference in mirror temperature (Martin et al., 2020).

Therefore there are some difference in how the HLOS data is assimilated in our Harmonie-Arome experiments. For the first period, with laser A in Sept-Oct 2018, we followed the recommendations from ECMWF (Rennie and Isaksen, 2020) and added a 1.35 ms$^{-1}$ bias correction to the Mie data. Further we added upper limits to the size of the observation error that was

acceptable to the data assimilation system. These were set to 4.5 ms$^{-1}$ for the Mie data and 8 ms$^{-1}$ for the Rayleigh data. The input data was also limited to one orbit per assimilation window corresponding to the orbit which had the most observations over the MetCoOp domain.

Between our first and our second experiment period, the Aeolus satellite switched the active laser, from laser A to laser B, on the onboard instrument because of the decrease in the laser energy for laser A. The switch took place in June 2019 (Martin

et al., 2020).

For the second period, with the laser B, the data available for the Mie was of a higher resolution (12 km horizontal distance rather than 90 km as was the case for the laser A period for both the Mie and Rayleigh data). Following our own experience with using Aeolus HLOS data (see section 5.1), we also decided to inflate the observation of the Mie data with a factor of 1.25 and also add a lower acceptable limit on the observation error so that all observations with an error lower than 1.5 ms$^{-1}$

were adjusted upwards to have an observation error of 1.5 ms$^{-1}$. The upper limit of the observation error of the Mie data was also slightly adjusted and set to 5 ms$^{-1}$ rather than 4.5 ms$^{-1}$. For the Rayleigh data all observation errors below 1 ms$^{-1}$ were set to 1 ms$^{-1}$ and the upper limit of the observation error for the Rayleigh data was kept at 8 ms$^{-1}$. Also worth mentioning is that since the ECMWF model, which assimilates Aeolus observations, is used to provide LBCs there can be a small additional benefit from Aeolus data improving the wind forecast in the LBCs for the Harmonie-Arome model runs.

The available Aeolus overpasses for the full experiments are shown in 1, where the laser A coverage is shown in the left hand panel and the laser B coverage is shown in the right hand panel. The orbits during the laser A period are more irregular than during the laser B period. The higher density of observations, particularly for the Mie observations, during the laser B period is also seen by the much smaller gaps between the available observations. For both periods, the 06 UTC is the one that has the most Aeolus observations over the MetCoOp domain.

In order to have a general idea of the performance of the Aeolus HLOS winds over the MetCoOp domain, we studied the difference between the observed values and the model background, which in this case is the three hour forecast from the previous cycle. In Fig. 2 we show the observation minus background (O-B) statistics, bias and standard deviation (STDV), for both types of HLOS winds compared against the other two sources of wind speed in the upper atmosphere in the experiment, radiosondes and aircraft data. The bias is close to zero for all the data types in both periods. The only exception being the



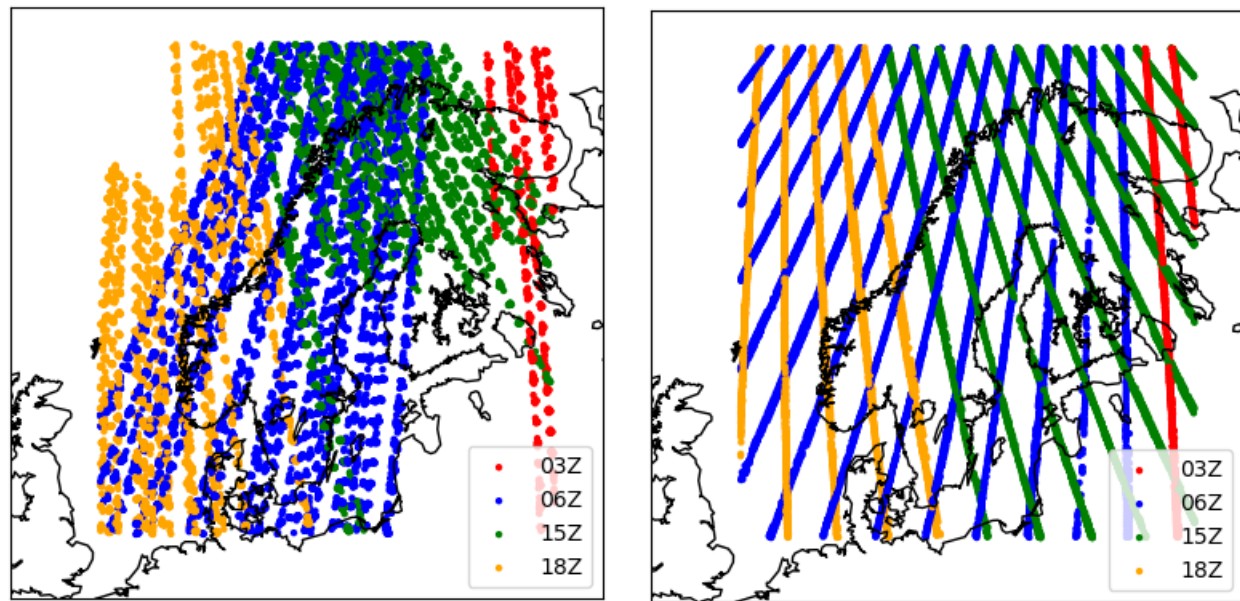

**Figure 1.** Available Aeolus overpasses during the experiment periods, for laser A (left) and laser B (right). The colour indicate the time of the overpass.

Rayleigh observations slightly below 400 hPa, which is likely caused by an undetected "hot pixel". The standard deviation show a larger discrepancy between the Aeolus data, both against each other and for the two different periods. Starting with the laser A period, we can see that the Mie data is showing similar values to the aircraft and radiosonde data with a standard deviation near 2 ms$^{-1}$. The Rayleigh data, as expected, show a larger standard deviation of around 4 ms$^{-1}$. For the laser B period, both the Mie and Rayleigh data quality has worsened. The standard deviation in the Mie data has doubled from 2 ms$^{-1}$

to 4 ms$^{-1}$, which is comparable to the Rayleigh values in the laser A period. The standard deviation of the Rayleigh data has also increased by nearly 2 ms$^{-1}$ and there are also larger fluctuations in the vertical profile. This reduced quality of wind data from laser B is caused by the laser's returning energy signal which was decreasing for this period.

During these two periods there is also a discrepancy in the availability of upper air wind observations. For the laser A period, Aeolus observations (both Mie and Rayleigh) correspond to 14% of the total wind observations in the upper air with the rest of

the data coming from radiosondes (35%) and aircraft measurements (51%). For the laser B period, in the spring of 2020 there were considerably fewer aircraft observation available, due to the limited number of flights during this period because of travel restrictions brought in as a measurement against the Covid-19 situation. For this period there are as many Aeolus observations as there are aircraft observations, both types of data make up 37 % of the total number of observations and radiosonde data make up the remaining 26% of upper air wind observations.

In the early period of the Aeolus satellite there were reports of discrepancy in the bias depending on the direction of travel of Aeolus (Rennie and Isaksen, 2020; Martin et al., 2020). Over the MetCoOp domain the HLOS observations available at 03





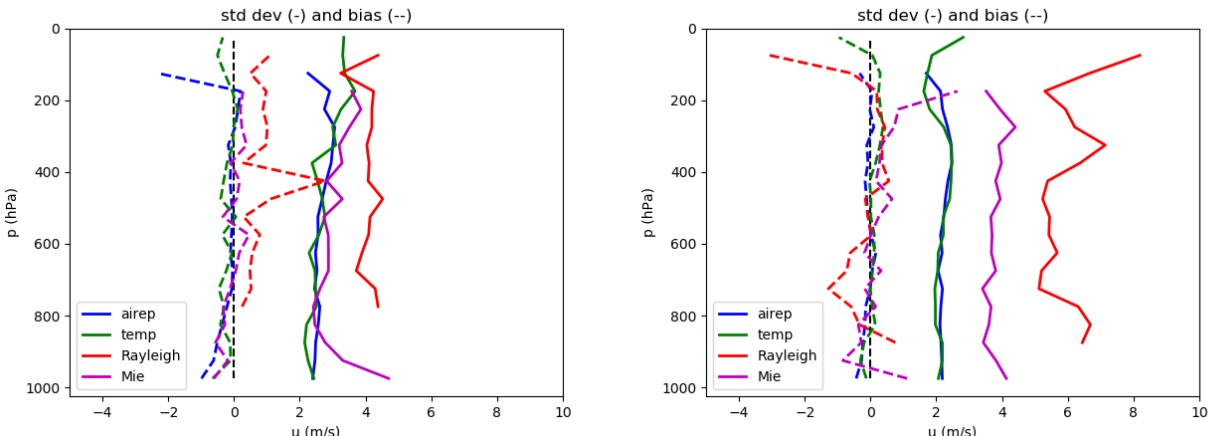

**Figure 2.** The standard deviation and bias of the observation minus background (O-B) for the laser A (left) and laser B (right) periods of Aeolus HLOS data (Rayleigh in red lines and Mie in magenta lines) against aircraft (blue lines) and radiosonde data (green lines).

and 06 UTC come from the descending part of the orbit (the satellite is travelling southward) and the observations at 15 and 18 UTC are from the ascending phase of the orbit (satellite travelling northwards). In Fig.3 we show the bias of the O-B values for ascending and descending orbits for both periods for both Mie (left) and Rayleigh (right). The top row show the O-B statistics

for the laser A period (Sept-Oct 2018), where for the bias a clear difference between the descending and ascending orbits can be seen in both Mie and Rayleigh data. There is a much smaller difference in the standard deviation where the Mie data has the smaller standard deviation for ascending orbits whereas for the Rayleigh data the descending orbits have a smaller standard deviation. The same comparison for the laser B period (Apr-May 2020) does not show the same difference in bias depending on the direction of travel of the satellite, except for the very lowest level of Rayleigh data. The differences in standard deviation

are also much lower, particularly for the Rayleigh data.

## 4   Impact in the NWP system

### 4.1   Impact on analyses

A first step to understanding the impact the Aeolus data has on our NWP system is to compare the observation minus background (O-B) statistics with the observation minus analysis (O-A) values. Figure 4 shows the difference in standard deviation

of the O-B and O-A for both types of HLOS data and for both periods. There is a clear difference between the O-B and the O-A standard deviation with height for both Mie and Rayleigh data, meaning that the Aeolus observations have an impact in the upper-air initial states of the NWP system. The Mie data has a larger impact within the data assimilation when estimating the initial state than the Rayleigh data for both periods, despite there being overall fewer Mie observations available. The larger



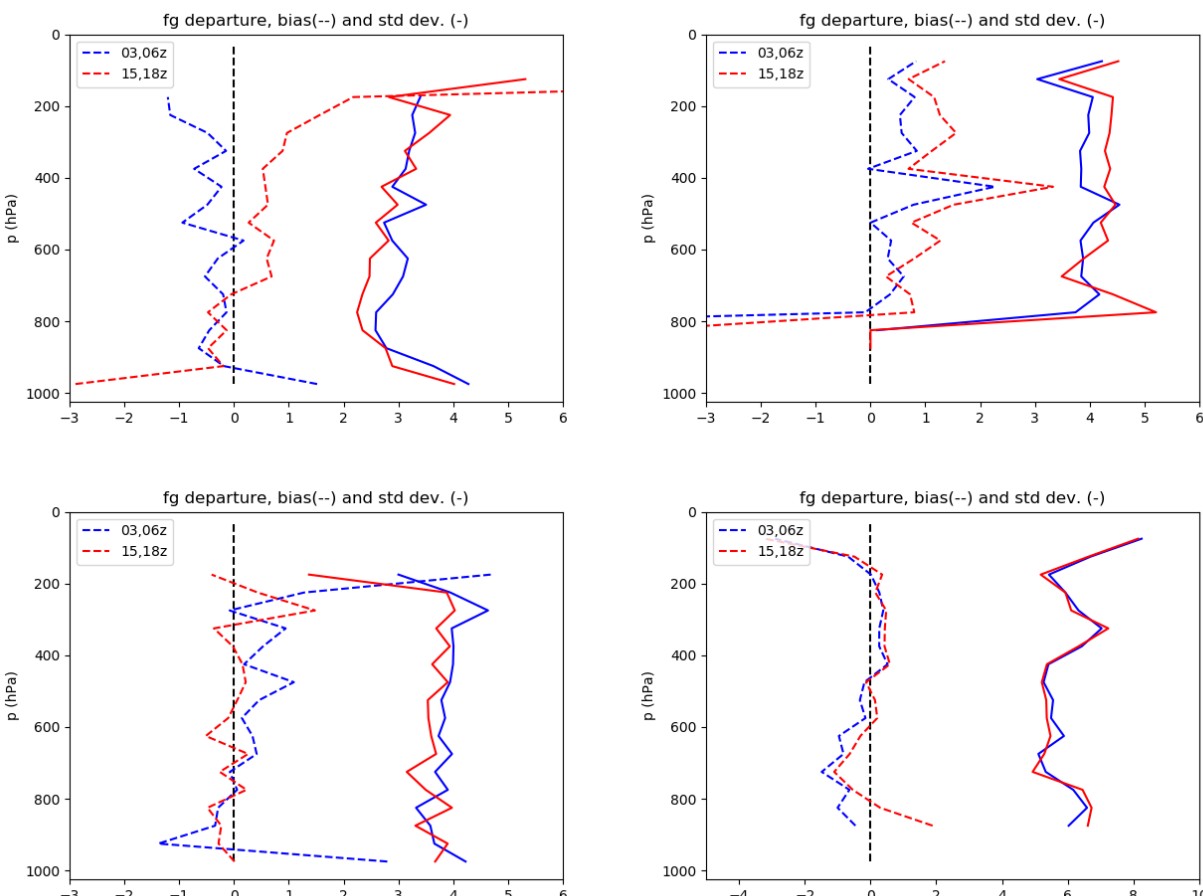

**Figure 3.** The bias and standard deviation of the observation minus background (O-B) for the laser A (top) and laser B (bottom) periods of Aeolus HLOS data. The Mie data in the left hand column and Rayleigh data in the right hand column. Blue lines mark ascending orbits and red lines descending ones.



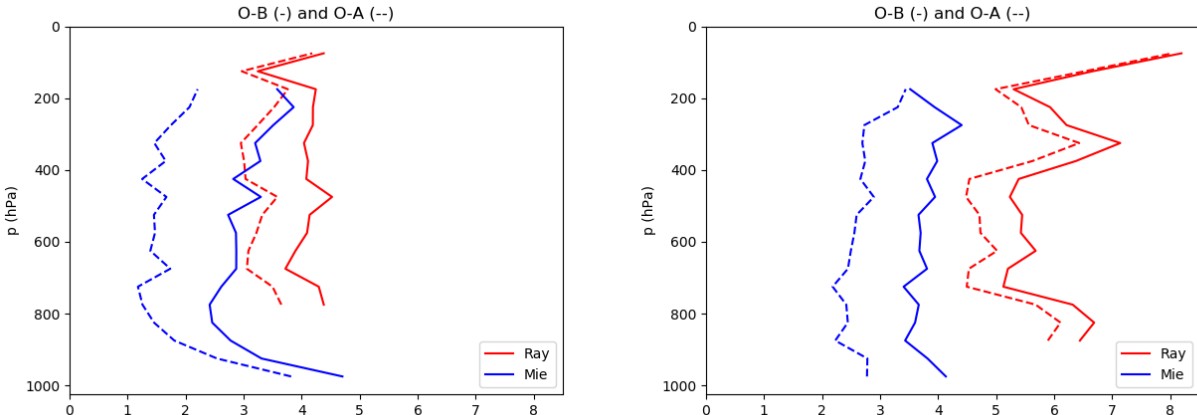

**Figure 4.** O-B (solid lines) and O-A (dashed lines) for Mie (blue) and Rayleigh (red) observations during the laser A period (left) and laser B period (right).

O-B values observed for the laser B period (see Fig 2) are also reflected in the larger O-A values for this period when compared
to the laser A period.

Another way to investigate the impact of the Aeolus data on the Harmonie-Arome model initial state is to calculate the Degree of Freedom of Signal (DFS). DFS is the derivative of the analysis increments in observation space with respect to the observations used in the analysis system and can be calculated using a randomization technique as proposed by Chapnik et al. (2006). This has the advantage as compared to O-B and O-A departure statistics that the total impact of all Aeolus HLOS
observations on the initial state can be estimated and compared with the impact of other observation types. The O-B and O-A departure statistics do not take the amount of observations into account. This is similar to relative DFS, which measures the impact on initial state per observation. The absolute DFS on the other hand represent the information brought into the analyses by the different observation types, in terms of amount, distribution, instrumental accuracy and observation operator definition. It provides information of the weight given to all observations of one particular type within the analysis system. There is also a
possibility of estimating the DFS per observation through calculation of relative DFS, by normalizing the absolute DFS by the amount of the observations belonging to one particular type of observations (Randriamampianina et al., 2011). The information obtained with relative DFS is comparable with O-B and O-A statistics.

Figure 5 show the DFS of all wind speed observations used in the data assimilation, SYNOP, radiosonde and aircraft data as well as Aeolus data. The figure to the left shows the DFS for all Aeolus data, the middle one for the Mie data and the right
hand one for Rayleigh data. This DFS calculation is done using all cycles of the day and every fifth day of the experiment period, which for the figure shown is the laser B period. The reason for using cycles from every fifth day is to use independent weather situations in the DFS statistics. The DFS clearly shows that for the Aeolus data the relative impact (bottom row) is significantly larger than the absolute impact (top row). The relative DFS shown in the lower panels of middle and right





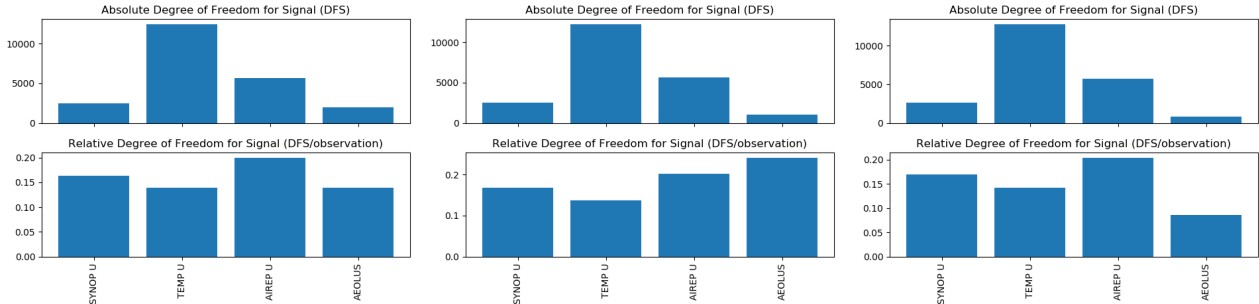

**Figure 5.** Degree of Freedom of Signal (DFS) for the experiment with all Aelous data (left), only Mie data (middle) and only Rayleigh data (right) for the laser B period compared with other sources of wind speed data. The absolute DFS is shown in the top row and the relative DFS in the bottom row.

columns show, consistently with Figure 4, that larger weight is given to Mie observations than to Rayleigh observations. From the corresponding upper panels it is evident that despite fewer observations, Mie observations have larger influence on initial state than Rayleigh observations. Furthermore, for the Mie only experiment, the Mie data has the largest relative impact of all of the observations. The Rayleigh data on its own also have a larger relative than absolute impact in the DFS values, but its relative importance is the smallest of the four observation types instead of the largest. When using all Aeolus data, the relative DFS of the Aeolus data is equal to that from the radiosonde data, which has the largest absolute DFS value.

If we repeat the DFS calculation, but only use the cycles for the same set of days where there are Aeolus observations (not shown), the absolute DFS for Aeolus increases. In two cases (all Aeolus data and Mie only) it has the third largest DFS of the four observation type. For the Rayleigh only experiment, the absolute DFS remains the smallest. For the relative DFS, the pattern seen using all cycles remains and the Mie only experiment is the only experiment where the relative DFS for Aeolus is the largest.

## 4.2 Impact on forecasts

To verify the wind speed forecast, we compare them to radiosonde data. These are available twice per day and in the MetCoOp domain we can find up to 18 radiosonde stations. Because the different availability times of the radiosonde and Aeolus HLOS observations, only some of the forecasts which are analysed in this section contain any direct assimilation of Aeolus HLOS data. In order to see as much impact of the Aeolus data as possible, we verify the forecasts after 6 hours, so that some of the forecasts will have used Aeolus data in the data assimilation.

The error standard deviation of the wind speed and direction for the 6 h forecasts for both periods for four different set ups (No Aeolus data (Ref), Mie only, Rayleigh only and both Mie and Rayleigh) is shown in Fig. 6. For the laser A period we can see a small improvement in the STDV of the wind speed for the Mie only experiment below 800 hPa. At 800 hPa there is also a small improvement in the STDV of the wind direction. For the laser B period the verification shows the worst wind speed STDV values for the Mie only experiment, whereas the best STDV for the wind direction below 500 hPa is also seen in the Mie





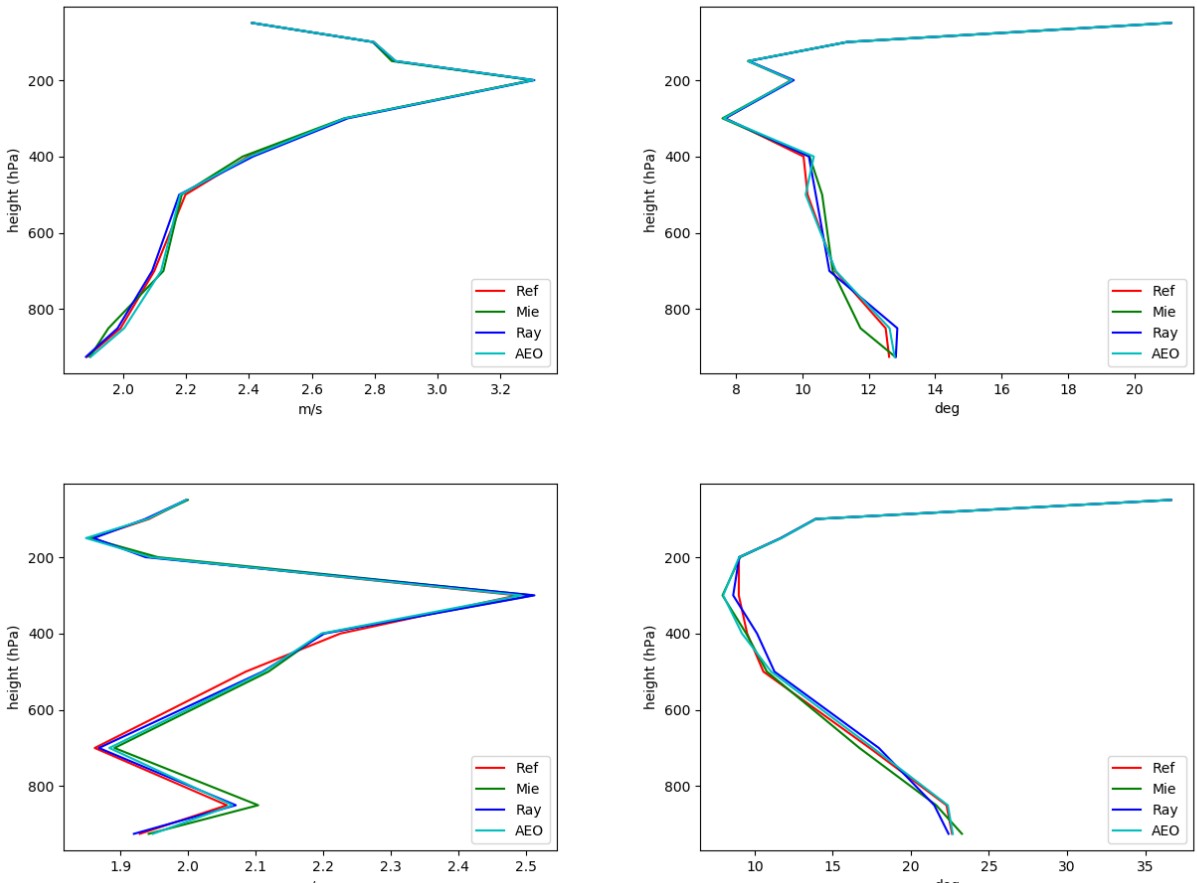

**Figure 6.** The error standard deviation of the wind speed (left) and wind direction (right) for the laser A period (top) and laser B period (bottom) for the 6 h forecast valid at 00 and 18 UTC. The experiment without Aeolus data is shown in red, the Mie only in green, the Rayleigh only in blue and the experiment using both Mie and Rayleigh in cyan. Beware the different sizes of the x-axis.

only experiment. Overall the impact of using Aeolus data in the verification is mostly neutral. The impact on other variables, such as temperature and pressure, is neutral for all experiments.

The bias of the wind speed for the 6 h forecast for this set of experiments (not shown) is around -0.2 ms$^{-1}$ for all experiments and periods with the largest bias (-0.4 ms$^{-1}$) found at 400 hPa height. The differences in bias between the different experiments is very small and it varies with height which experiment shows the lowest value of the bias. In general the wind speed bias is between 0 and -0.4 ms$^{-1}$ for laser A and between 0.1 and -0.5 ms$^{-1}$ for laser B. The bias for the wind direction varies with height and a negative bias is found near the surface and for the higher vertical heights.

We also looked at the O-B statistics for aircraft data for the cycles where Aeolus data was used by the background forecast to investigate the quality of the three hour forecast. As expected, the large-scale mixing have smoothed out the results and the





resulting STDV of O-B is neutral when comparing the Mie only and Rayleigh only experiment to the control forecast (without Aeolus data in the assimilation). This investigation was only conducted for the laser B experiment, but we anticipate that it would give a similar result for the laser A data.

## 5    Potential for enhanced use of Aeolus data

### 5.1    Tuning of error statistics in the present assimilation system

Following the method described by Desroziers et al. (2005), we analysed the observation and background errors for the experiment actively assimilating the Aeolus HLOS winds. The result is shown by the solid lines in Figure 7. This shows that the wind speed values given by the model are given more weight than the wind speed values from Aeolus, since there is a smaller error assumed for the model values. The estimation of the optimal values of these two parameters using the Desroziers method shows that while the model values should still be more trusted than the observations, both of them are trusted too much

and should be given less weight since the estimated values of $\sigma_B$ and $\sigma_O$ are larger than the ones that are used by the model. The importance of a proper representation of $\sigma_B$ values in limited area model data assimilation has earlier been studied by Lindskog et al. (2006). They found a positive impact on average verification scores, and that in addition a substantial positive impact is demonstrated for an individual synoptically active case by adjusting the observation error.

    A new experiment was run where the background error is increased (the model is less trusted) [1] and the observation errors

were also changed. Previous experiment used the observation errors as reported in the Aeolus data itself. Since the Desroziers diagnostic also indicated that the observations were given a too much weight, we manually changed all observation errors under 1.5 ms$^{-1}$ so that they were considered by the data assimilation system to have a 1.5 ms$^{-1}$ observation error. In another experiment we tested manually lowering the upper observation error as well, but this experiment showed us that the higher observation errors should be trusted and that artificially lowering them decreased the performance of the model.

The new values of $\sigma_B$ and $\sigma_O$ are shown on the left hand side of Figure 7 in dashed lines, comparing them to the original values (in the same colours but with solid lines) and we have increased the error more for the background than for the observations because the $\sigma_O$ will also have an impact on the other observations used in the data assimilation and the goal is also that the ratio of $\sigma_B$ and $\sigma_O$ of the used values should be close to what is estimated by the Desroziers diagnostics. Below 400 hPa this is achieved (blue dashed line in Fig. 7-right), but between 400 and 100 hPa, the ratio of the updated settings is smaller than

the recommended value.

    An experiment using both Mie and Rayleigh data as well as the updated settings was run for the full laser A period and it resulted in similar O-B as the reference experiment, but smaller O-A values (not shown), by on average 0.45 ms$^{-1}$ STDV. This means that overall the observations have been given more weight in the data assimilation and influence the analysis more. Looking at the verification scores, the impact of the change in background and observation error settings is neutral.

---

[1] This is done by changing the value of the REDNMC variable, which is set to 0.6; a decrease from its original value of 1.0.





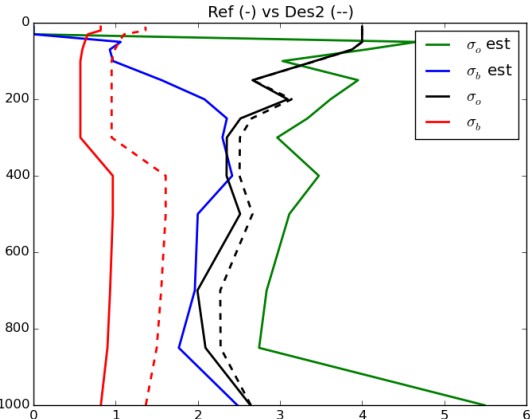
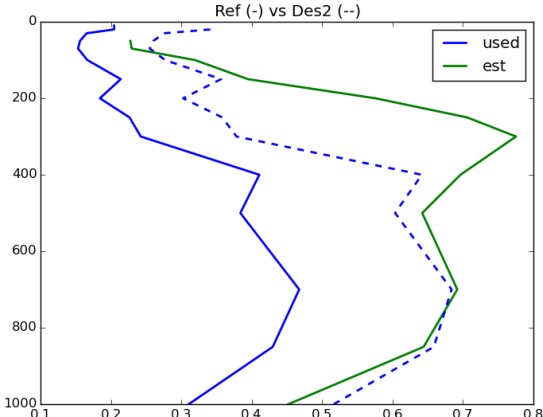

**Figure 7.** Left - The mean standard deviation of the background error ($\sigma_B$, red line) and the observation error ($\sigma_O$, black) used by the model and the estimated values of $\sigma_B$ (blue) and $\sigma_O$ (green) from the Desroziers diagnostics. The settings for the updated model are drawn with dashed lines. Right - The ratio $\sigma_B/\sigma_O$ between what was used (blue solid line) and the estimated ratio (green solid line) and the ratio used in the experiment with updated settings (blue dashed line).

In order to investigate whether there was a difference between the two types of HLOS observations, we ran the Desroziers diagnostic for the two experiments assimilating only Mie or only Rayleigh winds. These showed that the Desroziers diagnostic recommended a larger increase in the $\sigma_O$ for the Mie only experiment than what was shown by the same diagnostics when run for the Rayleigh only experiment. Taken these results into account when running the laser B experiments, we decided to inflate the observation errors for the Mie observations.

We also ran the Desroziers diagnostics for the laser B period for the experiment assimilating both types of Aeolus data. Also for this period the Desroziers diagnostic recommended that both the background and the observation errors should be increased; though both of the suggested increases are smaller than what was recommended for the laser A period. It is worth noting that the recommend increase in the background error was smaller even though the experiment was running with the default setting for the background error. It is easier to understand why the increase recommended for the observation error was smaller, since this was already increased for the Mie observations in the laser B experiment (see section 3).

### 5.2 Refined data assimilation technique

The potential of an enhanced use of Aeolus HLOS wind observation by application of an enhanced data assimilation technique has been exploited. This was achieved within a single observation data assimilation framework where the currently used 3D-Var method was compared with a 4-dimensional variational data assimilation (4D-Var, Courtier et al. (1994)) framework. Two main advantages of 4D-Var as compared to 3D-Var are that observations are used at their appropriate time and that (a simplified version of) the forecast model is used when minimizing the penalty function (Gustafsson et al., 2012). The latter imply a flow





dependency of data assimilation corrections of the background which has clear potential advantage for Aeolus HLOS data assimilation. As described by Gustafsson et al. (2018) data assimilation is less developed for km-scale models than for the meso-beta scale models. Nevertheless, a 4-dimensional variational data assimilation methodology has been developed for the Harmonie-Arome km-scale forecasting system and has the potential to further enhance the use of observations.

The single Aeolus HLOS observation parallel 3D-Var/4D-Var experiment was designed for the data assimilation cycle at 20200525 06 UTC. 3D-Var is designed to have a 3 h data assimilation time-window extending from 04.30 to 07.30 UTC. The present 4D-Var version is designed to have a 2 h data assimilation window starting at 05.00 UTC and ending at 07.00 UTC. The background state for 3D-Var is a 3 h forecast produced from a Harmonie-Arome initial state valid at 20200525 03 UTC. The background state for 4D-Var is a 2 h forecast launched from the initial state valid at 20200525 03 UTC. The 3D-Var initial state is produced at 06.00 UTC by minimizing a penalty function. The equivalent 4D-Var initial state at 06.00 UTC is produced by generating an initial state valid at 05.00 UTC, followed by a non-linear propagation of the increments to 06.00 UTC. From the 3D-Var and 4D-Var the generated initial states valid at 06.00 UTC, forecasts can be launched. The single simulated Aeolus HLOS observations is at 20200525 06.50 UTC. It is located at latitude, longitude 65.3°, 15.0° and at a vertical level of 679 hPa. The simulated Aeolus HLOS observation was from an ascending satellite orbit (North to South) and with instrument looking in a direction approximately towards the west. With this configuration a westerly observed wind gives negative Aeolus HLOS observation (positive direction defined to be for wind direction away from the lidar and negative direction is towards the lidar). Here the observed value is -7.0 ms$^{-1}$, and with an assigned observation error standard deviation of approximately 0.7 ms$^{-1}$. This corresponds to an accurate Mie Aeolus HLOS observations.

The 3D-Var background wind field at model levels around 2000 m as well as the wind and temperature fields at 300 hPa are shown together with the location of the single Aeolus HLOS observation in Fig. 8. A frontal structure is evident along the Swedish-Norwegian border with sharp gradients and an approximate North-South flow along the frontline. The single observation (marked with a dot) was positioned slightly east of the frontal area and valid at 50 min after the valid time of the 3D-Var background state.

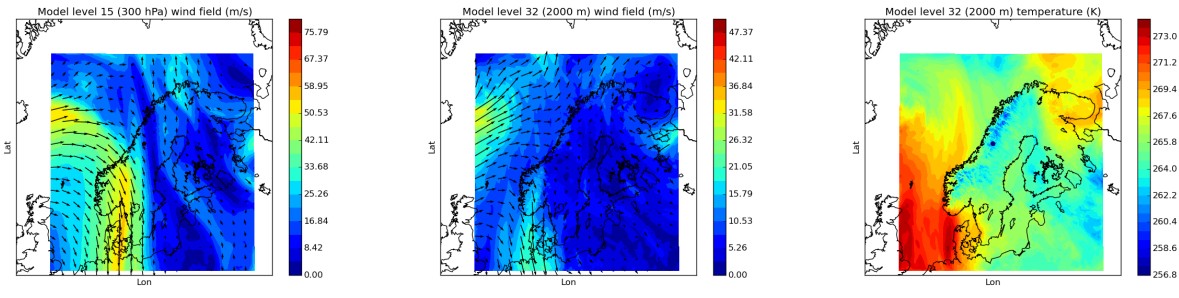

**Figure 8.** Wind (left and middle) and temperature (right) field of +3h forecast launched from 20200525 03 UTC and valid at 20200325 06 UTC. Also shown ii the horizontal location of single Aeolus HLOS observation with dot.



315 In Figure 9 the horizontal wind field assimilation increments at 06 UTC for model level 32 (around 2000 m) induced by the single Aeolus HLOS simulated observation is shown for 3D-Var (left) and 4D-Var (right). Important differences between 3D-Var and 4D-Var induced increments are that 3D-Var increments have considerably larger magnitude than the 4D-Var increments. Furthermore, the 4D-Var increments have smaller spatial scale and they are more flow dependent than the 3D-Var increments. This flow dependency is evident due to small scale variations due to the flow over a mountainous region and also

320 due to a more North-South component of the 4D-Var increments in agreement with the flow of the background state. This enhanced flow dependency is due to the utilization of the forecast model within the 4D-Var assimilation procedure. The main reason for the large difference in magnitude of assimilation increments between 3D-Var and 4D-Var is that with 4D-Var the observation is compared with a background model equivalent valid at 06.50 UTC, while with 3D-Var the observation is compared with a model equivalent valid at 06 UTC. In this highly flow dependent case with large wind increments this will result

325 in 3D-Var observation minus background departures of -2.3 ms$^{-1}$ and in 4D-Var observation minus background departures of -0.5 ms$^{-1}$.

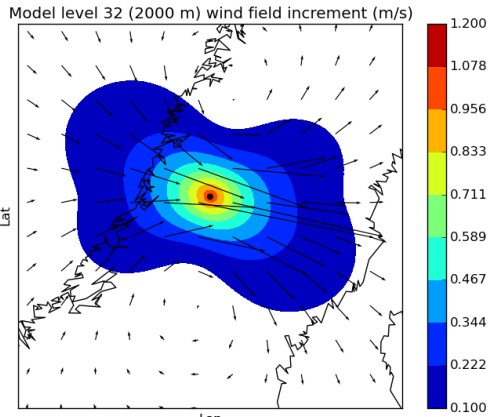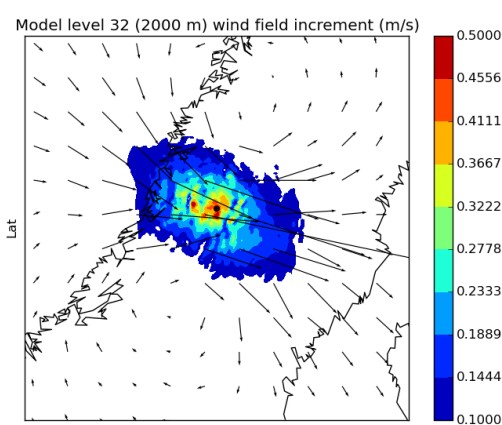

**Figure 9.** Model level 32 (around 2000m) 3D-Var (left) and 4D-Var (right) based analysis increments at 20200525 06 UTC, resulting Aeolus HLOS observation (position marked with a dot).

The idealized Aeolus HLOS single observation study has demonstrated that there is a clear benefit in the use of Aeolus HLOS observation from application of more flow dependent and advanced assimilation techniques taking the exact time of the observations better into account and it also makes better use of the model within the data assimilation process.

330 ## 6 Conclusions

Aeolus HLOS wind profiles have been added to the 3D-Var data assimilation in a regional high-resolution NWP system. In this study we have used the Harmonie-Arome model running over the MetCoOp domain covering the Nordic countries. We have



used the model to investigate the quality of the Aeolus satellite winds for two different four week periods, one early in the life of the satellite in Sept-Oct 2018 and the other after more than two years of operations in April-May 2020 and the impact of the Aeolus data on the Harmonie-Arome model has been investigated.

We conclude that Aeolus Mie data is demonstrated to be of considerably higher quality and more suitable for assimilation in our regional km-scale forecasting system than Rayleigh data. The Mie data are of a quality comparable to radiosonde and aircraft observations for the laser A period while Rayleigh has a lower quality. For the laser B period, even though the Mie quality is lower with respect to the laser A period, it still has the same relative higher quality compared to the Rayleigh data.

We have shown that the Aeolus data has an impact in the forecast and analysis as seen in the difference between O-B and O-A STDV profiles as well as through DFS analysis. From the DFS analysis we see that the Mie data has a larger impact on the analysis than the Rayleigh data and that the relative impact of the Aeolus data is larger than the absolute impact. Given that the information content from the Aeolus data, as seen in the total DFS, is small relative to the other sources of upper air data it's not unexpected that the Aeolus data has a mostly neutral impact on the verification of forecasts, with some small improvements seen in the wind speed and direction at selected height intervals. For the laser A period we noticed somewhat better verification scores if using Mie data only. This result is consistent with the analysis of the quality of Aeolus data.

Different approaches for further enhancing the impact of Aeolus data by new adoptions of the Harmonie-Arome data assimilation system were investigated. Such enhancements concerned tuned error statistics and application of a refined flow dependent assimilation technique. Results from analysis of error statistics based on Desroziers approach indicated that potential optimisations of currently used error statistics concerning Aeolus could be carried out. Application of such modifications resulted in rather neutral impact on forecast quality. A clear potential was however seen using a more refined assimilation techniques, in this case 4D-Var, allowing for taking the actual time of the observation better into account as well as the forecast model itself within the assimilation process.

All in all, we have found that the Aeolus data overall have a small impact on the Harmonie-Arome forecast. We have also seen that the impact of using the same LBCs for all experiments as well as the large-scale mixing scheme will have had an impact in smoothing out the Aeolus impact in our system. However, there are positive changes to the forecast and analysis which can be seen in the O-A, O-B statistics. As proposed by Stoffelen et al. (2020), having several Doppler wind lidar instrument in orbit and thus more overpasses would be more beneficial as we would have more data available and, potentially, for all our forecast cycles.

In the future, it would also be interesting to have a look at the reprocessed data for the laser A period and rerun these experiments in order to conduct a deeper analysis of the impact of the resolution, both vertical and horizontal, versus observation quality in our regional model. It would also be interesting to more fully examine the potential improvement in impact of Aeolus data on both analyses and forecasts if 4D-Var data assimilation is used for a longer trial. In addition it would be fruitful to coordinate the study with an ECMWF Aeolus experiment, to fully exploit the impact of Aeolus from regional data assimilation, LBC and large scale mixing. Moreover, in this study we mainly focused on observation error standard deviations, whereas potential observation error correlations would be a subject for future studies.



*Author contributions.* S. Hagelin is the main author and, together with R. Azad, responsible for running the experiments and performing data analysis. R. Azad also provided input on the manuscript and contributed to the 4D-Var experiment. M. Lindskog is responsible for the 4D-Var experiments and wrote part of to the manuscript. H. Schyberg and H. Körnich contributed to the writing of the manuscript.

*Competing interests.* We have no competing interests

*Acknowledgements.* SMHI was funded by Rymdstyrelsen (Swedish National Space Agency) Dnr 279/18. Met Norway was funded by by the ESA (European Space Agency) PRODEX (PROgramme de Développement d'Expériences scientifiques) programme.



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
