# Peer review of "Evaluating the use of Aeolus satellite observations in the regional NWP model Harmonie-Arome"

_Atmospheric Measurement Techniques, 2021_

## Referee Comment (RC1)

**Review of the paper „Evaluating the use of Aeolus satellite observations in the regional NWP model Harmonie-Arome"**

Dear authors,

first of all I think it is a very interesting paper concerning the impact of Aeolus wind lidar observations in a regional model for higher northern latitudes, where the impact of Lidar wind profiles show a substantial impact in global models. So far, the impact of Aeolus wind lidar observations in global models is documented very well, but the impact in regional models is not so well investigated. You nicely pointed out, that there a differences in quality between laser A and laser B and that the results are somehow independent of the observation error used as long as it substantially higher than the pure instrumental error. You also showed nicely that the use of flow dependent analysis due to the use of a 4-DVAR would help to benefit more from the Aeolus wind observations as this is visible in a 3DVAR system.

In the following I have only some small suggestions for some minor revisions.

I)     In row 17 of your introduction you write about AMVs derived from satellite radiances. Can you specify that a little clearer, For example "AMVs, derived from tracking cloud and water vapour image sequences.

II)    In row 28 I think it would be better to wright shift in the backscatter signal from the onboard laser.

III)   In row 30 the satellite did not observe the wind speed. Maybe it is better to wright "wind speed derived from the satellite measurements" is perpendicular etc.

IV)    In row 67 the Fig. numbering is ?? and the following sentence is not clear.

V)     Between row 100 and row 103 there is a double sentence

VI)    In row 112 you write that you use the same LBC date for both experiments. I don't understand that. I thought, that you use the ECMWF boundary data for Laser A were the ECMWF has not assimilated the AEOLUS data and for the second period the ECMWF assimilate the Aeolus date. That is a a major difference. Don't you think?

VII)   In row 142 you said that the horizontal distamce is 90 km and 12 km. in raw 125 you write 80 km an 10 km. What is correct ?

VIII)  In raw 142 you inflate the observation of Mie data but I think you probably inflate the observation error ot he Mie winds ?

As a final remark. I recommend to introduce a small section describing in a list or so all our experiments you have done (Crtl, All winds, only Mie winds, only Rayleigh winds) It seems to me this would give more structure and clarity in our paper.

---

## Referee Comment (RC2)

Review of the AMT manuscript: "Evaluating the use of Aeolus satellite observations in the regional NWP model Harmonie-Arome" by Hagelin et al.

**General comments.**

The manuscript describes the results of assimilating Aeolus Doppler wind lidar data in a regional high-resolution data assimilation system covering the Nordic region. It is an interesting and important study that complements a number of global NWP assimilation studies of Aeolus data by other scientists. Few, if any, similar studies have been performed, so it is certainly an area of research where publications are welcome in AMT and other journals. Before publication can be considered the manuscript needs to be tidied up and written with a more accurate use of data assimilation terminology. I have proposed a rather long list of related corrections below.

The conclusions are generally supported by the data. But the manuscript overstates the conclusions related to forecast and analysis impact evaluation from this study. The analysis increments and DFS are not good measures of forecast and even analysis impact, because they can be made artificially large by reducing the observation error. Too low observation error would give negative forecast impact. This would show up in forecast impact measures like 12-24-hour observation-minus-forecast statistics or FSOI (Forecast Sensitivity Observation Impact) statistics. The manuscript does include verification against radiosondes, that shows neutral results. So, due to the above the impact part of the conclusions has to be toned down.

It is a bit unclear what quality control has been applied to the data, both for the data assimilation process and for the diagnostics shown in Figs. 2-4. Are data in certain pressure ranges rejected? It is also unclear if observation errors have been inflated. Please improve the description of this in the manuscript.

In several places "model" is used where "data assimilation system" or "NWP system" is more appropriate. The data is not assimilated in the model. For example, I would prefer to replace "NWP model" with "NWP system" in the title. Please check and rewrite this throughout the manuscript. I have corrected some of the occurrences in my list of proposed corrections below.

The Aeolus data availability and usage within the region is a bit unclear: Is there only data available for the 03, 06, 15 and 18 UTC cycles? Why is the data limited to one orbit per assimilation cycle (line 136)?

If Aeolus data only is available for the 03, 06, 15 and 18 UTC cycles, it would be interesting to focus more on the statistics and impact for these cycles only. For Figure 5 it would very interesting to add the DFS statistics just for these four cycles, either in the form of a parallel set of bars or use of two colours to enable this in Figure 5.

It is useful to include Fig. 4, but it should be made clear in the text that STDV(O-B)-STDV(O-A) is not a measure of impact. It just reflects the observation error settings. Overfitting data would "look good" but give degraded forecast impact.

It would be informative to add curves with mean statistics for specified observation error to Fig. 4.

Lines 195-225: Be careful not to overestimate the use of DFS as an impact measure. DFS is very sensitive to observation error specifications. DFS does not estimate forecast impact. Please tone down the text accordingly.

It is not always clear what data assimilation experiments were performed: Did you run separate assimilation cycles with Rayleigh data and Mie data, respectively? I don't think this is necessary in order to get DFS and Desroziers diagnostics (line 276). Also, did you only use 4D-Var for the single observation experiments?

Conclusions lines 336-346: I think the statements are too strong and not justified based on the results. Aeolus Mie data will be more accurate than Rayleigh data due to Rayleigh-Brillouin scattering (Gu and Ubachs (2014), J. Chem. Phys.), but Mie data will have limited availability (only where there is clouds or aerosols). So, it is not so simple to determine if Mie or Rayleigh data is more useful for assimilation in regional km-scale models. Departures for Rayleigh data could be computed using model values averaged over several point to improve departure computations. This also relates to my comment above about the limitations in using increments and DFS to evaluate forecast impact. Basically, I don't think you can justify making such clear and definite statements about Mie versus Rayleigh data. So please tone down the text and remove unsubstantiated impact claims.

**More detailed comments**.

Line 14: A better data assimilation related reference would be: "*Some aspects of the improvement in skill of numerical weather prediction. A. J. Simmons and A. Hollingsworth (2002) QJRMS https://doi.org/10.1256/003590002321042135*"

Line 20: I suggest to update to a newer WMO reference. This one would be better: "*Fifth WMO Workshop on the Impact of Various Observing Systems on NWP (Sedona, Arizona, USA, 22 - 25 May 2012) - Workshop Report (Edited by: Erik Andersson and Yoshiaki Sato)*"

Line 23-24: Note, AMVs are also available from tracing moisture features. Please include radiosonde wind profiles in the list here.

Line 78: "applying ECMWF global EDA forecasts as lateral boundary conditions." Do you mean EDA (Isaksen et al. 2010) or EPS (Buizza et al. 2007)?

Line 78: "Ensemble Data Assimilation experiments (EDA)" → "Ensemble of Data Assimilations experiments (EDA, Isaksen et al 2010)"  That is if you meant EDA and not EPS.
                                              The refence is Isaksen et al. (2010) ECMWF RD TM 636.

Line 84: Are you averaging model fields along the 86 km BRC for Rayleigh data or using one model value for departure calculations?

Line 91-92: I find it easier to understand the equation in the form:
 $(H(x)-y)^2/(\sigma_b^2+\sigma_o^2) > L$
But you decide that.

Figure 2 and 3: Fig. 2 shows surprisingly low STDV(O-B) for Mie in Sep-Oct 2018. The values look smaller than for the similar data shown in Fig. 3 (left panel). Has it to do with using different quality control settings? Please check this.

Line 164: As noted just above, I'm surprised to see STDV(O-B) for Mie as low as 2 m/s for Sep-Oct 2018. Please check. The surprisingly large increase between first and second period for Mie departures is partly to reduced effective laser performance, as you mention, but also due to the change in accumulation length from 86 km to 12 km for Mie. I suggest to mention this in the manuscript.

Line 192: Based on my general comments, I suggest to rewrite "The Mie data has a larger impact within the data assimilation when estimating the initial state than the Rayleigh data" → "The Mie data, with its smaller observation errors, adjust the initial state more than the Rayleigh data"

Line 228 and 230: "only some of the forecasts which are analysed" and "we verify the forecasts after 6 hours, so that some of the forecasts will have used Aeolus data in the data assimilation"
It is unclear to me what you mean. Aeolus data is used in the appropriate 3-hour data assimilation cycle and not in the forecast. Please rewrite.

Figure 6: It would be most more useful to plot the difference between each experiment and the control experiment. This can either be in m/s or relative difference. Please consider to do this.

Line 250: Desroziers (2005) diagnostics is useful, but it provides an estimate with limitations. E.g., it assumes background error and observations error covariances are correctly specified. Please make this clear in the text.

Lines 270-275: It is unclear to me what you have done and why. Please improve the explanation.

Lines 280-285: It is difficult to understand what you have done. Please rewrite the text.

Line 307: The assigned observation error of 0.7 m/s seems very low. Please check and explain.

Figure 8 and the associated text lines 310-315: Please be consistent and either use pressure or altitude to describe level 15 and level 32.

Lines 326-329: It is useful to include Fig. 9 and the related text. But I believe it is an overstatement that the idealized single observation study demonstrated clear benefit of flow dependent 4D-Var versus 3D-Var. I believe it could well be true, but it cannot be concluded from these experiments. So, please tone down the text.

**Minor comments and proposed corrections.**

Line 31: Remove "most impacted by winds in"

Line 33: "used in clear" → "made in clear"

Line 43: It is "European Centre for Medium-Range Weather Forecasts"

Line 46: "models" → "NWP systems"

Line 48-49: Move LAM acronym from line 49 to line 48.

Line 54: "modelling" → "data assimilation"

Line 55: "observation and by model" → "observations and by the model"

Line 56: Remove "in some aspects"

Line 58: "observations." → "observations, respectively."

Lines 64 and 68: "3 h" → "3-hour"  ;  "1 h" → "1-hour"

Line 67: "Fig. ??" → "Fig. 1"

Line 74: "MetOp1 and 2" → "MetOp-A and MetOp-B"          (If this is what you mean)

Line 75: Remove "so called"  and "ideas of"

Line 94: Remove "so called"

Line 101-102: I suggest to remove the sentence "The Aladin … quite good."

Line 103: "On a later stage in the mission" → "In June 2019"

Line 103: "to improve the data quality and we" → "because the laser A had degraded in data quality. We …"

Line 105: "with correction for the M1 mirror temperature bias (Martin et al, 2020)."  → "with M1 temperature based bias correction (Rennie and Isaksen 2020)."

Line 106: "available and the data should have" → "available, so the data had"

Line 107: "is" → "was"

Line 116: "experiment run the model every three hours with 3D-Var" → "experiments used three-hourly 3D-Var"

Line 117: "catch-up runs of with a" → "only a"

Line 121: "models" → "systems"

Line 123: "80" → "86"

Line 125: "10 km after …" → "12 km. This was implemented …"

Line 125: "avoid the systematic error" → "account for Rayleigh-Brillouin scattering"

Line 131: Remove "orbital"

Line 131: "Martin et al. 2020" → "Rennie and Isaksen, 2020"

Line 134-135: "added upper limits to the size of the observation error that was acceptable to the data assimilation system. These were …" → "rejected poor quality data with large observation errors. These limits were …"

Lines 138- 142: "Between … Rayleigh data)."   Repetition of text above. Please remove.

Line 143: "observation" → "observation errors"

Lines 147-149: "Also … runs."   Repetition of text above. Please remove.

Line 150: "1" → "Fig. 1"

Line 152: "particularly for the Mie" → "due to the data sampling reduced from 86 km to 12 km for Mie"

Figs. 1-4 and 6: Please add the experiment periods to the legend is all these plots.

Line 160: "Likely caused" → "caused"

Line 160: After "hot pixel" add reference to (Fig. 8 in Rennie and Isaksen, 2020; Weiler et al., 2020). *Weiler, F., Kanitz, T., Wernham, D., Rennie, M., Huber, D., Schillinger, M., Saint-Pe, O., Bell, R., Parrinello, T., and Reitebuch, O.: Characterization of dark current signal measurements of the ACCDs used on-board the Aeolus satellite, Atmos. Meas. Tech. Discuss. [preprint], https://doi.org/10.5194/amt-2020-458, in review, 2020.*

Line 184: "depending on the direction of travel of Aeolus" → "for ascending and descending orbits"

Fig. 3: Please add a note in the legend that the bottom right panel is using different scaling.

Line 193: I suggest to remove: ", despite there being overall fewer Mie observations available."

Fig.4: It would be very useful to add curves with mean observation error as function of altitude for Mie and Rayleigh to the two panels.

Line 199: I suggest to remove: "as compared to O-B and O-A departure statistics"

Line 200-202: Please remove repeated text: "The O-B and O-A departure … per observation."

Line 202: Remove "on the other hand"

Line 208: "SYNOP" → "screen-level winds"

Fig. 5: Please improve figure: Same plot title is used on all panels in each row. This is not informative. Either remove or specify Absolute/Relative "All, Mie, Rayleigh DFS", as appropriate.

Line 216: "Furthermore, for the Mie only experiments, the" → "The"

Line 217: "The Rayleigh data on its own also have a larger relative than absolute impact in the DFS values,"   I don't understand what you mean.

Line 220: I would like you to, in Figure 5, also present the statistics just for the cycles with Aeolus data. I mentioned that above.

Line 236: I suggest to remove "in the verification"

Line 243: "used by the background forecast"  Do you mean "used by the analysis"?

Line 246: It would make sense to perform the investigation both for the laser A and the laser B experiment, and combining the statistics to ensure more reliable results. Please consider to do this.

Line 252: "wind speed values given by the model are given more weight than the wind speed values from Aeolus, since there is a smaller error assumed for the model values." → "wind speed background errors (red solid lines) are smaller than wind speed observation errors for Aeolus (black solid lines)."  The original text is confusing.

Line 254: "model values" → "background"

Line 259: "model" → "background"

Line 264: "performance of the model" → "forecast skill"

Line 264: "Figure 7" → "Fig. 7"

Line 271: "using both Mie and Rayleigh data"
Was that not the case for the case for the experiments above. Please clarify.

Figure 7 legend: "model" → "data assimilation system"  two times!

Line 283: "recommend" → recommended"

Line 300: "launched" → "produced"

Line 301: "produced" → "computed"

Line 303: "initial states" → "analyses"

Line 315: "Figure 9" → "Fig. 9"

Line 327: "demonstrated" → "indicated"

Figure 9: Add to legend "Note, different colour scales used for the two panels"

Line 333: "model" → "assimilation system"

Line 335: "model" → "NWP system"

Line 339: Please mention the reduction of averaging length scale from 86 km to 12 km for Mie as a partly explanation for this.

Line 352: "a more refined assimilation techniques in this case 4D-Var" → "a 4D-Var assimilation technique"

Line 352: "as the" → "as using the"

Lines 354-358: I don't think the manuscript has shown what is written about LBC and positive forecast impact. I suggest to remove "We have also … O-A, O-B statistics."

Line 360: "have a look at" → "use"

Line 371: "by by" → "by"

Line 394:  Write out the list of coauthors.

Line 403: "and K., M."  - Please update reference

Line 412: Improve Pourret reference and add doi-link

Line 430: "Quartely" → "Quarterly"  !!

---

## Author Comment (AC1)

Response to Reviewer 1

We would like to thank the reviewer for their kind words. We have addressed all points raised by the reviewer in the manuscript and you can see our specific comments to them here below.
Best regards,
the authors

I) In row 17 of your introduction you write about AMVs derived from satellite radiances. Can you specify that a little clearer, For example "AMVs, derived from tracking cloud and water vapour image sequences.
Suggestion added to the manuscript.

II) In row 28 I think it would be better to wright shift in the backscatter signal from the onboard laser.
Suggestion added to the manuscript.

III) In row 30 the satellite did not observe the wind speed. Maybe it is better to wright "wind speed derived from the satellite measurements" is perpendicular etc.
Suggestion added to the manuscript.

IV) In row 67 the Fig. numbering is ?? and the following sentence is not clear.
There was a typo in the latex-reference for the figure number. It has now been corrected. Likewise there was a missing word in the sentence, which has now been added, making it less clear than it should have been.

V) Between row 100 and row 103 there is a double sentence
Double sentence is now removed.

VI) In row 112 you write that you use the same LBC date for both experiments. I don't understand that. I thought, that you use the ECMWF boundary data for Laser A were the ECMWF has not assimilated the AEOLUS data and for the second period the ECMWF assimilate the Aeolus date. That is a a major difference. Don't you think?
Yes, this is correct. For the laser A period there are no Aeolus data in the ECMWF boundary data, whereas for the laser B period, ECMWF do assimilate Aeolus data. For this study, all model runs within the same period use the same data as LBCs. What we would have liked to do, as is explained here, is to run a further set of experiments with our regional model, one experiment using LBC data with Aeolus data in the assimilation and one experiment with LBC data where the Aeolus data is not used in the assimilation. If we had this set of experiments we would have a better idea of what the total influence of the Aeolus data in our model and what is the impact is from the LBCs and what is the impact from the model's own assimilation of Aeolus.

VII) In row 142 you said that the horizontal distamce is 90 km and 12 km. in raw 125 you write 80 km an 10 km. What is correct ?
The reference we used for these numbers stated the distances as 86.4 km for Rayleigh and 12 km for Mie. In places in this manuscript we have rounded these numbers, we have changed these and use 86 and 12 km throughout the text now.

VIII) In raw 142 you inflate the observation of Mie data but I think you probably inflate the observation error ot he Mie winds ?

Indeed, that is what we did. Thank you for spotting it. The missing word has now been added.

As a final remark. I recommend to introduce a small section describing in a list or so all our experiments you have done (Crtl, All winds, only Mie winds, only Rayleigh winds) It seems to me this would give more structure and clarity in our paper.
We have added a list at the end of section 2 to clarify which experiments we run.

---

## Author Comment (AC2)

Reviewer 2

We wish to thank the reviewer for their very thorough review. We have done our best to update and improve the manuscript in line with the suggestions made. We respond to each of the issues raised here below under each item.

Review of the AMT manuscript: "Evaluating the use of Aeolus satellite observations in the regional NWP model Harmonie-Arome" by Hagelin et al.

General comments.

The manuscript describes the results of assimilating Aeolus Doppler wind lidar data in a regional high-resolution data assimilation system covering the Nordic region. It is an interesting and important study that complements a number of global NWP assimilation studies of Aeolus data by other scientists. Few, if any, similar studies have been performed, so it is certainly an area of research where publications are welcome in AMT and other journals. Before publication can be considered the manuscript needs to be tidied up and written with a more accurate use of data assimilation terminology. I have proposed a rather long list of related corrections below.

The conclusions are generally supported by the data. But the manuscript overstates the conclusions related to forecast and analysis impact evaluation from this study. The analysis increments and DFS are not good measures of forecast and even analysis impact, because they can be made artificially large by reducing the observation error. Too low observation error would give negative forecast impact. This would show up in forecast impact measures like 12-24-hour observation-minus-forecast statistics or FSOI (Forecast Sensitivity Observation Impact) statistics. The manuscript does include verification against radiosondes, that shows neutral results. So, due to the above the impact part of the conclusions has to be toned down.

It is a bit unclear what quality control has been applied to the data, both for the data assimilation process and for the diagnostics shown in Figs. 2-4. Are data in certain pressure ranges rejected? It is also unclear if observation errors have been inflated. Please improve the description of this in the manuscript.

The Aeolus data is quality controlled by using the observation error available in the observation files produced by ECMWF and data with too high observation error are rejected. This is described in lines 132-137 for the laser A period and lines 141-147 for the laser B period. (line numbers refer to the original submission.)

In several places "model" is used where "data assimilation system" or "NWP system" is more appropriate. The data is not assimilated in the model. For example, I would prefer to replace "NWP model" with "NWP system" in the title. Please check and rewrite this throughout the manuscript. I have corrected some of the occurrences in my list of proposed corrections below.

We respectfully disagree to some extent. Personally, I tend to think of all parts of an NWP system as a model rather than as data assimilation, forecasting model, post processing and so on, but some of my co-authors agree with you. We have updated the manuscript were these changes have been suggested in the text but prefer to keep the title.

The Aeolus data availability and usage within the region is a bit unclear: Is there only data available for the 03, 06, 15 and 18 UTC cycles?

Correct, though for on any particular day we usually have data from two or three of these cycles. Depending on the day, we sometimes only have one cycle with Aeolus data and sometimes (though very rarely) all four.

Why is the data limited to one orbit per assimilation cycle (line136)?

At the start of the laser A experiments we thought that it would be better to use the observations closest to analysis time in our 3D-Var system, but we later understood that using more observations matter even more, so all available orbits were used for the laser B experiments.

If Aeolus data only is available for the 03, 06, 15 and 18 UTC cycles, it would be interesting to focus more on the statistics and impact for these cycles only. For Figure 5 it would very interesting to add the DFS statistics just for these four cycles, either in the form of a parallel set of bars or use of two colours to enable this in Figure 5.

We have this data too. Though the result is very similar to the all cycles data, and given that we don't have Aeolus data for ALL of the cycles where we potentially can have Aeolus data, we felt it more useful to show all cycles.

This is what the DFS looks like if it's calculated for the 03, 06, 15 and 18 UTC cycles (left: all Aeolus, middle: Mie only, right: Rayleigh only). The largest change is that aircraft data now has a larger absolute DFS than radiosondes and the absolute DFS for Aeolus is somewhat higher. The relative DFS for Aeolus is very similar to that we see in Fig. 5.

[Figure]

It is useful to include Fig. 4, but it should be made clear in the text that STDV(O-B)-STDV(O-A) is not a measure of impact. It just reflects the observation error settings. Overfitting data would "look good" but give degraded forecast impact.

We have added a sentence to make this clearer and we completely agree. That was one reason for applying Desroziers statistics to obtain realistic error specifications.

It would be informative to add curves with mean statistics for specified observation error to Fig. 4.

We have added a second row to figure 4, which shows the average observation error for Mie and Rayleigh data. We have furthermore, for the sake of completeness decided to introduce background error equivalents, as estimated from the horizontal wind component background error. The ratio between background and observation error equivalents for, Mie and Rayleigh observations respectively, influence the departure between O-B and O-A departure statistics illustrated in upper panel.

Lines 195-225: Be careful not to overestimate the use of DFS as an impact measure. DFS is very sensitive to observation error specifications. DFS does not estimate forecast impact. Please tone down the text accordingly.

We describe what the DFS statistics show here. We have tried to tone down the text elsewhere in the manuscript.

It is not always clear what data assimilation experiments were performed: Did you run separate assimilation cycles with Rayleigh data and Mie data, respectively? I don't think this is necessary in order to get DFS and Desroziers diagnostics(line 276). Also, did you only use 4D-Var for the single observation experiments?

We have added a bullet list with the experiments we have run for both periods. We hope that this will provide the needed clarity. Yes, we did run separate assimilation experiments with just Rayleigh data and another with just Mie data. And yes, we only used 4DVar for the single observation experiment. We would like to run further experiments with 4DVar as we believe this will allow us to see a larger impact of using Aeolus data.

Conclusions lines 336-346: I think the statements are too strong and not justified based on the results. Aeolus Mie data will be more accurate than Rayleigh data due to Rayleigh-Brillouin scattering (Gu and Ubachs (2014), J. Chem. Phys.), but Mie data will have limited availability (only where there is clouds or aerosols). So, it is not so simple to determine if Mie or Rayleigh data is more useful for assimilation in regional km-scale models. Departures for Rayleigh data could be computed using model values averaged over several point to improve departure computations. This also relates to my comment above about the limitations in using increments and DFS to evaluate forecast impact. Basically, I don't think you can justify making such clear and definite statements about Mie versus Rayleigh data. So please tone down the text and remove unsubstantiated impact claims.

We are aware of the smaller observation error for the Mie data in comparison with the Rayleigh data as well as the limitations in the availability of the Mie data. However, we still believe that in our case (in this specific region and for these two periods) we see more benefits (albeit small ones) in using Mie data

over using Rayleigh data. We think that more studies are needed, looking at other areas and time periods in order to make any definitive statements about the impact of Aeolus data in regional models in general. We are currently investigating the impact of Aeolus data in another region in the Arctic. We hope that this study will be of interested.

More detailed comments.

Line 14: A better data assimilation related reference would be: "Some aspects of the improvement in skill of numerical weather prediction. A. J. Simmons and A. Hollingsworth(2002) QJRMS https://doi.org/10.1256/003590002321042135"
Reference changed.

Line 20: I suggest to update to a newer WMO reference. This one would be better: "Fifth WMO Workshop on the Impact of Various Observing Systems on NWP (Sedona, Arizona, USA, 22 -25 May 2012) -Workshop Report (Edited by: Erik Andersson and Yoshiaki Sato)"
Thank you. Reference updated.

Line 23-24: Note, AMVs are also available from tracing moisture features. Please include radiosonde wind profiles in the list here.
Radiosondes are mentioned are mentioned on lines 20-21. The sentence regarding AMVs has been updated to clarify that AMVs track cloud tops.

Line 78: "applying ECMWF global EDA forecasts as lateral boundary conditions." Do you mean EDA (Isaksen et al.2010) or EPS (Buizza et al.2007)?
We mean EDA (Isaksen et al).

Line 78: "Ensemble Data Assimilation experiments (EDA)" → "Ensemble of Data Assimilations experiments (EDA, Isaksen et al 2010)" That is if you meant EDA and not EPS. The refence is Isaksen et al.(2010) ECMWF RD TM 636.
We are using the EDA. Text is updated with the Isaksen reference.

Line 84: Are you averaging model fields along the 86 km BRC for Rayleigh data or using one model value for departure calculations?
We treat it as a point based observation and then perform a horizontal and vertical interpolation.

Line 91-92: I find it easier to understand the equation in the form:$(H(x)-y)^2/(\sigma_b^2+\sigma_o^2) > L$ But you decide that.
Equation has been updates as suggested.

Figure 2 and 3: Fig. 2 shows surprisingly low STDV(O-B) for Mie in Sep-Oct 2018. The values look smaller than for the similar data shown in Fig. 3 (left panel). Has it to do with using different quality control settings? Please check this.
We have checked these, and the issue here is that the Mie curve is slightly, but steadily, increasing with height. When writing about figure 2 we focused too much on the values near 800 hPa, rather than noticing the tilt of the curve. Looking closer at Fig. 2, we believe that the true value for the Mie std deviation

is closer to 3 than 2 m/s. The different scale of the x-axis between figs 2 and 3 also added to the confusion. The mean value for the full curve is 3.13 m/s. The text has been updated accordingly.

Line 164: As noted just above, I'm surprised to see STDV(O-B) for Mie as low as 2 m/s for Sep-Oct 2018. Please check. The surprisingly large increase between first and second period for Mie departures is partly to reduced effective laser performance, as you mention, but also due to the change in accumulation length from 86 km to 12 km for Mie. I suggest to mention this in the manuscript.
We're sorry for the confusion. The lowest Mie value is 2.41 m/s, but it's mostly nearer to 3 m/s. We have added a sentence about the change in the accumulation length.

Line 192: Based on my general comments, I suggest to rewrite "The Mie data has a larger impact within the data assimilation when estimating the initial state than the Rayleigh data" → "The Mie data, with its smaller observation errors, adjust the initial state more than the Rayleigh data"
Sentence changed as suggested

Line 228 and 230: "only some of the forecasts which are analysed" and "we verify the forecasts after 6 hours, so that some of the forecasts will have used Aeolus data in the data assimilation"It is unclear to me what you mean. Aeolus data is used in the appropriate 3-hour data assimilation cycle and not in the forecast. Please rewrite.
Sorry for the confusion. Of course the Aeolus data is used in the data assimilation. We have tried to clarify this in the manuscript.

Figure 6: It would be most more useful to plot the difference between each experiment and the control experiment. This can either be in m/s or relative difference. Please consider to do this.
We also consider it useful to know the overall error in m/s and since the forecast impact is neutral we prefer to keep the figure as it is.

Line 250: Desroziers (2005) diagnostics is useful, but it provides an estimate with limitations. E.g., it assumes background error and observations error covariances are correctly specified. Please make this clear in the text.
We have added a sentence here to explain this.

Lines 270-275: It is unclear to me what you have done and why. Please improve the explanation.
We ran a further experiment with tweaked settings, but didn't think the results were noteworthy enough to be included in this paper with full figures and results, so we just added a brief discussion on the results, see lines 280-285 in the first submitted version. Some minor clarifications have been added to the text, which hopefully will make it this paragraph clearer.

Lines 280-285: It is difficult to understand what you have done. Please rewrite the text.
We have tried to make this section a bit clearer.

Line 307: The assigned observation error of 0.7 m/s seems very low. Please check and explain.
We did check again and this value is correct. We chose a data point with a low observation error in the processed Aeolus data in order to easier be able to see a clear impact of the observation in this demonstration study.

Figure 8 and the associated text lines 310-315: Please be consistent and either use pressure or altitude to describe level 15 and level 32.
Sorry, we are now consistently using meters as the height unit.

Lines 326-329: It is useful to include Fig. 9 and the related text. But I believe it is an overstatement that the idealized single observation study demonstrated clear benefit of flow dependent 4D-Var versus 3D-Var.I believe it could well be true, but it cannot be concluded from these experiments. So, please tone down the text.
As requested, we have toned this section down.

Minor comments and proposed corrections.

Line 31: Remove "most impacted by winds in"
Sentence is modified.

Line 33: "used in clear" → "made in clear"
text is changed

Line 43: It is "European Centre for Medium-Range Weather Forecasts"
Forecasting changed to Forecasts

Line 46: "models" → "NWP systems"
changed

Line 48-49: Move LAM acronym from line 49 to line 48.
moved

Line 54: "modelling" → "data assimilation
Change made.

Line 55: "observation and by model" → "observations and by the model"
"the" added

Line 56: Remove "in some aspects"
Removed

Line 58: "observations." → "observations, respectively."
Added "separately" instead of "respectively"

Lines 64 and 68: "3 h" → "3-hour" ; "1 h" → "1-hour"
This looks strange to me, I prefer to keep the text as is.

Line 67: "Fig. ??" ->"Fig. 1"
Corrected

Line 74: "MetOp1 and 2" → "MetOp-A and MetOp-B" (If this is what you mean)
Corrected

Line 75: Remove "so called" and "ideas of"
removed

Line 94: Remove "so called"
removed

Line 101-102: I suggest to remove the sentence "The Aladin … quite good."
Double sentence removed

Line 103: "On a later stage in the mission" → "In June 2019"
Changed

Line 103: "to improve the data quality and we" → "because the laser A had degraded in data quality. We …"
Changed

Line 105: "with correction for the M1 mirror temperature bias (Martin et al, 2020)." → "with M1 temperature based bias correction (Rennie and Isaksen 2020)."
Text changed

Line 106: "available and the data should have" → "available, so the data had"
sentence modified

Line 107: "is" → "was"
corrected

Line 116: "experiment run the model every three hours with3D-Var" → "experiments used three-hourly 3D-Var"
Sentence changed

Line 117: "catch-up runs of with a" → "only a"
changed

Line 121: "models" → "systems"
changed

Line 123: "80" → "86"
Changed

Line 125: "10 km after …"  → "12 km. This was implemented …"
changed

Line 125: "avoid the systematic error" → "account for Rayleigh-Brillouin scattering"
Systematic errors kept, but we also added the information about Rayleigh-Brillouin scattering.

Line 131: Remove "orbital"
We decided to keep it. The effect of the bias in our domain is a larger bias in the afternoon than in the morning orbits.

Line131: "Martin et al. 2020" → "Rennie and Isaksen, 2020"
Changed

Line 134-135: "added upper limits to the size of the observation error that was acceptable to the data assimilation system. These were ..." → "rejected poor quality data with large observation errors. These limits were ..."
sentence modified

Lines 138-142: "Between ... Rayleigh data)." Repetition of text above. Please remove.
Lines removed.

Line 143: "observation" → "observation errors"
Corrected

Lines 147-149: "Also ... runs." Repetition of text above. Please remove.
removed

Line 150: "1" → "Fig. 1"
corrected

Line 152: "particularly for the Mie" → "due to the data sampling reduced from 86 km to 12 km for Mie"
text modified

Figs. 1-4 and 6: Please add the experiment periods to the legend is all these plots.
Added for Fig. 1. For the rest of the figures we prefer to refer the reader to the text.

Line 160: "Likely caused" → "caused"
removed

Line 160: After "hot pixel" add reference to (Fig. 8 in Rennie and Isaksen, 2020; Weiler et al., 2020). Weiler, F., Kanitz, T., Wernham, D., Rennie, M., Huber, D., Schillinger, M., Saint-Pe, O., Bell, R., Parrinello, T., and Reitebuch, O.: Characterization of dark current signal measurements of the ACCDs used on-board the Aeolus satellite, Atmos. Meas. Tech. Discuss. [preprint], https://doi.org/10.5194/amt-2020-458, in review, 2020.
Both added

Line 184: "depending on the direction of travel of Aeolus" → "for ascending and descending orbits"
changed

Fig. 3: Please add a note in the legend that the bottom right panel is using different scaling.
Added

Line 193: I suggest to remove: ", despite there being overall fewer Mie observations available."
We decided to keep it.

Fig.4: It would be very useful to add curves with mean observation error as function of altitude for Mie and Rayleigh to the two panels.
Figure has been updated.

Line 199: I suggest to remove: "as compared to O-B and O-A departure statistics"
Removed

Line 200-202: Please remove repeated text: "The O-B and O-A departure ...per observation."
Sentences removed.

Line 202: Remove "on the other hand"
removed

Line 208: "SYNOP" → "screen-level winds"
As the wording SYNOP is also used in the figure, we prefer to keep it.

Fig. 5: Please improve figure: Same plot title is used on all panels in each row. This is not informative. Either remove or specify Absolute/Relative "All, Mie, Rayleigh DFS", as appropriate.
We have added All, Mie and Rayleigh in the appropriate location

Line 216:"Furthermore, for the Mie only experiments, the"→"The"
changed

Line 217:"The Rayleigh data on its own also have a larger relative than absolute impact in the DFS values," I don't understand what you mean.
We mean that the absolute DFS of the Rayleigh data in the Rayleigh only experiment is smaller than the relative DFS.

Line 220: I would like you to, in Figure 5, also present the statistics just for the cycles with Aeolus data. I mentioned that above.
See response above.

Line 236: I suggest to remove "in the verification"
removed

Line 243: "used by the background forecast" Do you mean "used by the analysis"?
Used for the data assimilation which produces the 3 h forecast which the analysis uses.

Line 246: It would make sense to perform the investigation both for the laser A and the laser B experiment, and combining the statistics to ensure more reliable results. Please consider to do this.

We prefer to keep the periods separate for the time being. It would make more sense to combine the data once the reprocessed data for the early laser A period is available.

Line 252: "wind speed values given by the model are given more weight than the wind speed values from Aeolus, since there is a smaller error assumed for the model values."->"wind speed background errors (red solid lines) are smaller than wind speed observation errors for Aeolus (black solid lines)." The original text is confusing.
Text corrected.

Line 254: "model values" → "background"
changed

Line 259: "model" → "background"
changed

Line 264: "performance of the model" → "forecast skill"
changed

Line 264:"Figure 7" → "Fig. 7"
changed

Line 271: "using both Mie and Rayleigh data" Was that not the case for the case for the experiments above. Please clarify.
Yes, that is the case. We wanted to emphasis that this particular rerun was only made for the case using all Aeolus data. We also ran the Desroziers diagnostics for the two experiments using only Mie and only Rayleigh, but found no results worth mentioning in the paper. We have removed the phrase in quotation marks noted above to avoid any confusion here.

Figure 7 legend: "model" → "data assimilation system"two times!
Changed, twice!

Line 283: "recommend" → "recommended"
changed

Line 300: "launched" → "produced"
changed

Line 301:"produced" → "computed"
changed

Line 303: "initial states" → "analyses"
changed

Line 315: "Figure 9" → "Fig. 9"
ok

Line 327: "demonstrated" → "indicated"
This section has already been modified as per the initial request to to tone down the findings a bit, and as part of that we already made this change.

Figure 9: Add to legend "Note, different colour scales used for the two panels"
Added

Line 333: "model" → "assimilation system"
changed

Line 335: "model" → "NWP system"
changed

Line 339: Please mention the reduction of averaging length scale from 86 km to 12 km for Mie as a partly explanation for this.
Mentioned

Line 352: "a more refined assimilation techniques in this case4D-Var" → "a 4D-Var assimilation technique"
changed

Line 352: "as the" → "as using the"
change added

Lines 354-358: I don't think the manuscript has shown what is written about LBC and positive forecast impact. I suggest to remove "We have also …O-A, O-B statistics."
One sentence removed and the other reworded

Line 360: "have a look at" → "use"
changed

Line 371: "by by"→ "by"
changed.

Line 394: Write out the list of coauthors.
Added. Too much copying and pasting on our part, and presumably wherever it was pasted from had a limit on how many co-authors it was possible to specify.

Line 403: "and K., M."-Please update reference
a comma was missing from the .bib file. This has now been corrected

Line 412: Improve Pourret reference and add doi-link
Improved, though it's too early to provide a doi-link at this stage

Line 430: "Quartely" → "Quarterly"!!
missing "r" added

---

## Author Response (AR2)

Response to report 1

I am very happy with the authors response to the reviewers comments. They have addressed all the issues we raised in an fully acceptable way. Congratulations to the authors for providing an excellent paper on use of Aeolus data in a high-resolution limited area model assimilation system. I would though propose to update figure 5 to include an extra row with DFS values for the cycles where Aeolus is available. The authors have already prepared such a figure in the response to me. Only the top row with absolute DFS is required to be added, as the relative DFS values are very similar to the ones presented. This would provide a fairer picture of the absolute DFS of Aeolus data versus other observing systems for the cycles where Aeolus data is available.

Thank you for your comments. We have added the absolute DFS for the hours were Aeolus are available to figure 5 and updated the manuscript accordingly.
Kind regards,
the Authors

Report 2

Review of Hagelin et al.
The manuscript title Evaluating the use of Aeolus satellite observations in the regional
NWP model Harmonie-Arome provides an investigation into the impact of Aeolus HLOS winds assimilation in the 3DVar Harmonie-Arome model over the MetCoOp domain. The paper describes well the datasets used from Aeolus, details about the implementation into the DA system, and impacts both the analysis and forecast. The result suggest that the Aeolus data has a positive impact on wind analysis, with Mie winds providing larger impact, but more neutral impact on the forecast. The reviewer recommends however, that more details be given on the forecast impact, by examining more metrics relevant for regional NWP, especially humidity and precipitation forecast skill which should be influenced by improve wind analysis. Therefore the reviewer recommends publication of the manuscript after major revisions.

Thank you for your comments. We have updated the manuscript to address the issues that have been raised by this review. Details are reported here below.
Kind regards,
the Authors

Minor Comments:
Page 1, Line 22: Atmospheric Motion Vectors is already defined- change "Atmospheric Motion Vectors (AMV)" to "AMVs"
Second definition removed.

Page 1, Line 22-23: AMVs can also be computed by water vapor image sequences as noted earlier in the text, so it is not correct to say wind speed is only measured at cloud top height, but also in layers of the atmosphere where satellite observations are sensitive to water vapor signal. I think the point still remains that vertical coverage with all AMVs is limited.
Sentence lightly modified to take this into account.

Page 2, Line 32: Suggest saying the wind measurement is usually near the zonal component of the wind vector rather than east-west since this changes depending on ascending/descending orbit.
Zonal added as suggested, but is east-west kept in parenthesis, as I feel this is easier to understand intuitively rather than zonal component (which I always need to look up to remember which direction is the zonal component and which is the meridional component).

Page 3, Paragraph 2: Could you please explicitly mention the horizontal spatial resolution of the Harmonie-Arome model and also the thinning grid resolution for Aeolus observations?

The resolution of the Harmonie-Arome grid is 2.5 km, as mentioned on line 66, and we don't thin the Aeolus observations. We calculated the horizontal correlations of the Aeolus data and concluded that there was no need to thin the data.

Page 4, Line 110: change "providing to sets of LBC" to "providing two sets of LBC"

Thank you for spotting this. A w is added in the appropriate place

Page 4, Line 114: change "All the experiment" to "All the experiments"

added

Page 4, Line 122: change "first satellite-based lidar mission" to first satellite-based wind lidar mission"

wind added.

Page 5, Line 141: Can you please define the MetCoOp domain? I do not see any description previously in the text.

Line 65 and onwards describe the MetCoOp domain briefly and the area covered can be seen in figure 1.

Page 5, Line 145: Can you describe the method used to conclude the inflation factor and the rationale for adjusting the observation error limits upward? Is this based on looking at O-A stats from the control, or some other method?

Following our previous experience of these observations, the results of the Desroziers diagnostics and the recommendations from ECMWF, we decided this would be an appropriate inflation factor.

Page 7, Figures 2/3: Is it possible to include plots of observation counts for the Mie and Rayleigh winds used to compute the statistics? And also mention average number of observations assimilated per 3 hour cycle.

We have added a black dotted line to each panel in figure 3 to show the number of Aeolus observations used in the data assimilation for the full period. The number of observations per 3 h cycle varies, in particular for the Mie observations, but on average for the laser B period we use ~200 observations for the Mie data and ~450 observations for the Rayleigh data. Since we only use one overpass for the laser A experiments, the figures are lower here, ~180 Rayleigh observations and ~60 Mie observation per cycle.

We did not add the observation count to figure 2, as that would create a very messy plot with too many lines. The information is the same as what is now shown by the black dotted lines in figure 3.

Major Comments:

Page 7, Figures 2/3 and text: Is there an additional bias correction used to remove the residual bias of the HLOS O-B illustrated for the Rayleigh winds for the laser A data? If not, can you please add text to highlight this point and any impact it might have on the analysis?

Just the regular observation error check. This bias will be removed in future reprocessed Aeolus datasets. At the time we did these experiment, such bias correction was not available. We have added a sentence on variation bias correction to the manuscript, to outline what can be done in case the reprocessed Aeolus data doesn't completely remove this bias over our domain.

The bias correction used in our studies is the same as that used by ECMWF where they estimated the bias after running experiments for long periods, but we used a shorter period both for laser A

and laser B. We would have implemented additional bias correction if we had the opportunity to experiment with longer periods.

Section 4.2: The section on forecast impacts should be expanded to include other metrics, particularly with the application in regional NWP. Realizing that the impacts on the wind forecast are relatively small in terms of speed and wind vector. It may be worth looking at the impact of the u and v wind components (perhaps more wind impact on the u component), and then assess other forecast metrics, especially precipitation or specific humidity as the transport of moisture variables should be impacted by Aeolus information added to forecast initialization.

Our current verification system only allows us to verify wind speed and direction, not the wind speed components themselves. Presumably we would see the same impact (possibly marginally larger) in the u component and no impact for the v component.

The radiosonde network over the MetCoOp region is rather sparse and it is difficult to properly verify small-scale parameters like humidity with this network. We do have verification of specific humidity and other parameters, but we decided not to include them as the impact of Aeolus data is negligible. We have added a sentence concerning this to the end of the section about the impact on forecasts.

The image below show the result for specific humidity when comparing against radiosonde data for the laser B period. The only visible result is a small degradation of the standard deviation when using the Rayleigh data at the lowest levels. The vertical profiles of specific humidity for the laser A period are even more neutral than for the laser B period.

For the precipitation, (shown on the next page) perhaps there is a small improvement in using the Rayleigh data, but it's not large enough to draw any conclusion other than that the Aeolus data have a neutral impact on the forecast scores. Looking at the same score for laser A, there is a corresponding tiny improvement in precipitation from using all Aeolus data and Rayleigh only is the worst performing.

[Figure]

No cases

---

## Author Response (AR3)

Revisions for
Evaluating the use of Aeolus satellite observations in the regional NWP model Harmonie-Arome

Minor revisions to the manuscript have been performed, increasing its scientific value . A few points for further consideration remain in my view.

Thank you for the comments. In particular, the last comment contributed to the improvement of the manuscript and has hopefully taught its main author to stop confusing speed with velocity, once and for all. The manuscript has been revised to address the issues raised here.
Best regards,
the authors

1) I'd be careful with statements about analysis impact and DFS, as these are related to error covariance settings and not one to one related to the additional information content brought by Aeolus. In particular, it would be useful to more explicitly stress that a large analysis impact is not necessarily a good thing, as it may lead to overfitting and detrimental effects in the subsequent forecast. Please check.

We have modified this section further to better reflect this issue.

2) In terms of data assimilation, global NWP is followed to a large extent, while particular considerations for mesoscale wind data assimilation in terms of spatial characteristics and sampling, which are very different for the Mie and Rayleigh channel, are not highlighted. For example, a recent publication from the HARMONIE community could be useful to discuss in this respect, i.e., https://doi.org/10.1002/qj.3979 and references therein.

We have added text, both to the main part of the paper and to the conclusions, concerning the use of super-modding.

3) In the introduction, wind speed is often highlighted, while the WMO OSCAR wind requirement is for the 2D horizontal wind component of the wind vector. This is much in line with the initialization of 3D turbulence characteristics in the atmosphere on the mesoscale, where the wind vector is relevant and its variability well expressed by its vector components, rather than wind speed and direction. It would be useful to rephrase the introduction with this requirement in mind in my view and avoid the abundant use of "wind speed". Furthermore, Aeolus measures one of the vector components and it would be useful to evaluate its impact in terms of vector components. I agree with reviewer 2 that speed and direction verification are less meaningful. If the capability is not present to track and verify vector component changes in the forecasts, it may be worthwhile to recommend this as a useful capability for the future in the manuscript.

There has been a misunderstanding from my part here. I misremembered and assumed that "wind speed" implied both the strength of the wind and its direction, rather than only the strength of the wind. I confused speed with velocity, velocity is the one that describes the vector wind and not the other way around! Wind speed used in the incorrect sense has been removed from the manuscript. Why I managed to use the correct terminology while writing the "impact on forecasts" section, but not in the rest of the manuscript is beyond my understanding.

---

## Author Response (AR4)

Response – revision 4

Comments to the Author:
Many thanks for these further improvements.

However, I note a new misquote on Mile et al., which also affects Mie wind assimilation. You write "In the case of observations representing courser spatial scales than the model, this scale difference has been demonstrated to be successfully handled by application of a so-called supermodding approach (Mile et al., 2021)."

Note that ASCAT winds have 25-km resolution, but Mile et al. find 60-km supermodding to work best. This is model aggregation beyond the observation resolution. Mile et al. show furthermore that supermodding changes the spatial scales that are extracted from the observations. The idea of supermodding is that those spatial scales are initialized that best improve the short-range forecasts. Initializing the small-scale noise of the model over the oceans and in the upper air is not thought to be productive. See for a more complete motivation:
https://nwp-saf.eumetsat.int/site/download/documentation/scatterometer/reports/ High_Resolution_Wind_Data_Assimilation_Guide_1.3.pdf , as referred to by Mile et al.

So, I'd suggest something like: "To deal with model noise and spatial representation of observations, a careful evaluation of data assimilation in terms of initializing targetted spatial scales needs further evaluation for the quiite different spatial characteristics of the Mie and Rayleigh winds. For example, model noise over the ocean has been demonstrated to be successfully handled by application of a so-called supermodding approach (Mile et al., 2021)."

Thank you for the clarification. The manuscript has been updated as suggested.
Best wishes,
the authors